# A Systematic Compilation of Human SH3 Domains: A Versatile Superfamily in Cellular Signaling

**DOI:** 10.3390/cells12162054

**Published:** 2023-08-12

**Authors:** Mehrnaz Mehrabipour, Neda S. Kazemein Jasemi, Radovan Dvorsky, Mohammad R. Ahmadian

**Affiliations:** 1Institute of Biochemistry and Molecular Biology II, Medical Faculty and University Hospital Düsseldorf, Heinrich Heine University Düsseldorf, 40225 Düsseldorf, Germany; mehrnaz.mehrabipour@hhu.de (M.M.); neda.jasemi@hhu.de (N.S.K.J.); 2Center for Interdisciplinary Biosciences, P. J. Šafárik University, 040 01 Košice, Slovakia

**Keywords:** proline-rich motifs (PRM), protein interaction, SH3 domain, SH3 domain-containing proteins, signal transduction, SRC homology 3

## Abstract

SRC homology 3 (SH3) domains are fundamental modules that enable the assembly of protein complexes through physical interactions with a pool of proline-rich/noncanonical motifs from partner proteins. They are widely studied modular building blocks across all five kingdoms of life and viruses, mediating various biological processes. The SH3 domains are also implicated in the development of human diseases, such as cancer, leukemia, osteoporosis, Alzheimer’s disease, and various infections. A database search of the human proteome reveals the existence of 298 SH3 domains in 221 SH3 domain-containing proteins (SH3DCPs), ranging from 13 to 720 kilodaltons. A phylogenetic analysis of human SH3DCPs based on their multi-domain architecture seems to be the most practical way to classify them functionally, with regard to various physiological pathways. This review further summarizes the achievements made in the classification of SH3 domain functions, their binding specificity, and their significance for various diseases when exploiting SH3 protein modular interactions as drug targets.

## 1. General Introduction

The SRC homology 3 (SH3) domain was first described in 1988 as a region of approximately 60 amino acids found in different intracellular signaling proteins, such as SRC and PLC [1,2]. SH3 domains are arranged as small protein modules in a compact β-barrel fold made of five β-strands connected by RT, n-SRC, distal loops, and a 3_10_-helix (Figure 1) [3]. Thousands of SH3 domains present in eukaryotes, prokaryotes, and viruses have been investigated and characterized as modules mediating the protein–protein interaction/association [4,5]. SH3 domain-mediated protein–protein interactions have significant diversification as the binding partners regulate almost all essential cellular functions, including cell survival, proliferation, differentiation, migration, and polarity. Moreover, findings underscore the significance of SH3 domains in shaping protein–protein interaction, their potential influence on protein folding and positioning, their impact on cellular phenotypes, and the essential role they play in protein function [6]. Mutations and malfunctions of the SH3 domain can lead to significant neurological defects, cancer, and infectious diseases [7,8,9].

SH3 domain-containing proteins (SH3DCPs) have a complex array of potential physiological partners due to their ability to recognize diverse structural scaffolds that are both dependent on, and independent of, the consensus proline-rich motif (PRM). This allows them to favor typical and atypical specific recognition sites. Biochemical and structural studies have been published on peptide libraries recognized by SH3 domains. These studies have been used to predict potential binding partners containing this sequence to gain a better understanding of SH3-mediated biological responses [10]. Human SH3DCPs represent a populous and well-characterized family with almost 300 domains embedded in 221 large multidomains and small monodomain proteins. A novel multidomain phylogenetic analysis of SH3DCPs shows their co-occurrence across a large set of protein domains, and it provides insight into their functional prerequisites in different signaling pathways. In this review, we focus on the specificity landscape underlying protein–protein interactions that are mediated by SH3 modules and the functional diversification of SH3 domains in human signaling pathways based on their phylogenies and relations to different diseases.

## 2. Specificity of Binding

SH3 domains, among other peptide-binding modules, provide multivalent binding by increasing the avidity of interactions and promoting phase transition during physical interactions with a pool of ligands called proline-rich motifs (PRMs) [11,12,13]. As certain interactions between the SH3 domain and PRMs are fundamental to the assembly of multiprotein complexes, it is reasonable to assume that the SH3DCPs are involved in a wide variety of cellular processes [14,15,16]. A set of five types of PRM-binding modules, including SH3, WW, EVH1, GYF, and UEV, have been reported to date [15,17,18,19,20]. PRMs are typically composed of proline (P) and hydrophobic (X) amino acids, with a core canonical motif XPxXP (where x can be any amino acid). The distinctive cyclic structure of proline’s side chain gives proline an exceptional conformational rigidity compared with other amino acids. This unique structural property of proline may interfere with the regular formation of secondary structures, making it more abundant in unstructured regions. Consequently, proline residues are frequently exposed on the surface of proteins, making them accessible for interaction with other proteins or molecules [21]. The outstanding feature of PRMs is the actual degree of combinatorial diversity, which is determined by the presence of one or more proline residues [22,23,24]. The PRMs can be classified into three different types, including short linear sequence motifs with prolines that are involved in protein–protein interactions, like canonical PxXP [25,26], tandem repeats containing multiple copies of the same motifs in a row, like the two adjacent PPII helical PxXP motifs involved in the interaction with IRTKS-SH3 [26], and clustered motifs, which are multiple copies of the same motif that are found near each other. An example of proline clustering is an assembly of synaptic vesicle proteins that are bound with *SH3DCPs* in nerve terminals [27].

A canonical SH3 domain interaction with proline-rich peptides (PRPs) is characterized by specific hydrophobic contact recognition and the interaction of positively charged PRP residues with negatively charged residues of the SH3 domain [24,28]. Additionally, there are also water-mediated hydrogen bonds contributing to binding that is crucial for the stabilization of complexes [29,30]. The spatial arrangement of conserved amino acids located close to each other on the surface of the SH3 domain presents the PRM binding surface. PRM binding occurs at three major sites, involving the hydrophobic patch (Tryptophan), which is flanked by the n-SRC loop, as well as the RT loop and β4-α3_10_ of the SH3 domain (Figure 1) [31,32]. The SH3 domain can bind to their binding partners in two opposite orientations, defined by the relative positioning of non-proline residues, which are mostly positively charged residues [32,33]. The location of this basic residue, designated as +x/x+, determines the orientation of peptide binding in relation to the conserved proline residues at the N-terminal (+xXPxXP, class I) or the C-terminal (XPxXPx+, class II) positions of PxXP core [26,34,35]. For all SH3 domains, Arg is the basic residue defining the orientation, aside from some exceptions wherein Lys is the flanking residue for the second SH3 domain of TSPOAP1, the first SH3 domain of CRK, and SH3 domain of CTTN [10,36]. In both classes I and II, the structural and mutational analysis and studies suggest that the SH3–PRM interaction can, after initial major binding recognition, engage flanked areas outside the proline-rich core which regulates and increases binding specificity [37]. A structural comparison of SH3 domain binding sites shows that the higher variability and flexibility of loop regions account for the specificity and affinity in PRP binding [38,39]. The selectivity of the SH3 domain in particular PRPs is generally modest, with affinities usually in the low micromolar range [23,31,32,33,40]. An example of class I is a complex between the SH3 domain of MYO1E through the N-terminal Arginine 358, and Prolines 371 and 374 in FAK [41]. The crystalline structure of the second SH3 domain of CD2AP in complex with Pro-457, Pro-459, and Arg-462 in RIN3, shows the preference for class II orientation [42]. Some SH3 domains can bind to either class I or class II categories; FYN-SH3 is one such example [34,43,44,45]. A comprehensive study on binding specificities for 115 SH3 domains has shown that numerous SH3 domains exhibit extended alternative selectivity to non-proline residues in a peptide motif [10]. A crystallography and isothermal titration calorimetry (ITC) study of GRAP2-SH3C (MONA) and GRB2-SH3N clearly shows an unexpected binding combination concerning the essential RXXK motif of HPK1, which complements the PxXP motif [46]. A micromolar range affinity has also been found between the SH3 domain of STAM2 and GRB2-SH3C with the PX(V/I)(D/N)RXXKP motif of UBPY and SLP-7, respectively [47,48]. Another consensus PXXDY sequence was identified in ABI1 (E3B1) and RN-tre, in which DY was found to be crucial for binding, and the proline residue provided considerable specificity for EPS8-SH3 [49]. Furthermore, NCK2-SH3.1 forms a connection with the unique PxxDY motif found in the cytoplasmic tail of CD3ε. This motif includes Tyr166 within the ITAM subdomain of CD3ε. By associating with this motif, NCK2-SH3.1 hinders the phosphorylation of Tyr166, subsequently regulating the activity of the T-cell receptor [50]. The N-terminal SH3 domain of NCK1, together with EPS8, is also verified to show specificity for the PxxDY motif [51].

SH3 domains in several studies recently discovered that SH3DCPs also exhibit an extended repertoire of binding sequences, known as proline-independent binding, allowing SH3DCPs to mediate a broader array of interactions [19]. An example of atypical binding is the SH3 domain of RASA1, the RAS-specific GAP (p120RASGAP), which interacts with the catalytic GAP and kinase domains of DLC1 and Aurora kinases, respectively, thereby inhibiting their activities [52,53]. Other findings demonstrate a specific Intramolecular interaction between the SH3 and Guanylate Kinase (GuaKin/GK) domain of DLG4 (PSD-95) that predominates over intermolecular associations. Unlike the typical binding of SH3 domains to poly-proline motifs, SH3/GK binding occurs through a bi-domain interaction that necessitates intact motifs [54]. As a non-traditional binding mode, the SH3 domain can also play a role in facilitating the formation of intricate scaffold structures. The binding of the SH3-SH3 domains in ITSN1 and SH3GL2 (endophilin1) leads to their association, and this complex is recruited to locations wherein the clathrin-mediated recycling of synaptic vesicles takes place. This association facilitates the uncoating of vesicles at neural synapses [55]. In another study, the five SH3 domains of ITSN1 are associated with the autoinhibition of the DH domain, indicating that the PxXP-binding groove on the SH3 domain does not play a role in this interaction [56]. Interestingly, SH3 domains can also be involved in RNA binding. According to a study from Pankivskyi et al. in 2021, the interaction between ITSN1-SH3D and mRNA promotes the solubilization of RNA-binding protein, SAM68. This occurs via interactions with ITSN1-SH3A and the mRNA-binding protein, SAM68-PRM; this triple complex may lead to the recruitment of specific mRNA for splicing regulation [57]. Other atypical interactions involve helix structures as interacting partners for *SH3 domains*. The C-terminal SH3 domain of NCF2 (p67phox) binds to the non-PxXP peptide segment of NCF1(p47phox) in helix–turn–helix arrangements [58]. Further research has indicated that non-PxXP alpha-helical motifs are essential and adequate for the binding of Pex5p to the PEX13-SH3 domain [58,59]. A notable feature of PEX13 is that it can simultaneously bind to both the canonical type II PRM sequence on Pex14p and the non-canonical binding site on Pex5p with a different binding surface on the SH3 domain [60,61]. In another study on *C. elegans* muscle, the interaction between UNC-89’s SH3 (homologs of human OBSCN-SH3) and coiled α-helical structures of paramyosin, which shares a strong homology with skip2 residues on human cardiac Myosin (MYH7), leads to the mislocalization of paramyosin [62]. In a separate investigation, it was discovered that the interaction between FYN-SH3 and the N-terminal “RKxxYxxY” motif of SKAP55 necessitates the presence of arginine and lysine residues [28]. This study found that the RKxxYxxY motif was also recognized by SH3 domains that can bind to canonical class I motifs, whereas class II SH3 domains, like GRB-2, were unable to do so [28]. However, it was also shown that GRB2-SH3_c-term_ and Gads can recognize and bind to an R-X-X-K motif of SLP-76 [63]. Moreover, the 40-fold difference in binding affinity for GRB2 suggests that GRB2-SLP-76 formation does not occur in vivo, in comparison with Gads, to facilitate receptor T cell signaling, suggesting that other factors are involved in mediating complex formation [48,63]. In another example, the interaction between BIN1-SH3 and its internal domain, referred to as Exon10, contains the basic sequence RKKSKLFSRLRRKKN, which hinders the SH3 domain from interacting with its typical PxXP ligand in dynamin [64]. Similarly, CdGAP activity is inhibited by the SH3 domain of ITSN1 by direct binding to its central basic-rich (BR) region comprising Lys and Arg residues (xKx(K/R)K motif) [65,66]. Another non-canonical binding of SH3 domains is found in the trinary SLAM–SAP–FYN-SH3 complex, in which the SAP-SH2 domain binds to FYN-SH3, thus linking FYN to SLAM immune receptors [67,68]. NCF1 (p47phox) also contains Arg70-Ile-Ile-Pro-His-Leu-Pro76, a canonical class I SH3 binding residue within the PX domain that can be recognized by its C-terminal SH3 domain; however, the surrounding PX structure also contributes to the production of a higher affinity [69]. The MACF1 protein belongs to the plectin family, which contains spectrin repeats (SR) and an SH3 domain in the middle, suggesting an SR4–SH3 interaction that stabilizes intermolecular contacts [70]. In addition, other domains, such as the LIM4 domain of PINCH-1, can also trigger rapid focal adhesion by transiently interacting with NCK2-SH3.3 [71]. In another interesting example of non-canonical binding, the single SH3 domain of CASKIN1 lacks key aromatic residues from the canonical binding groove, causing the protein to behave differently. There is a recent report suggesting that it might bind to membrane surfaces with high levels of LPA [72]. As with PRAM1-SH3, charged residues in the RT loop mediate a relatively high affinity for PI(4)P, and to a lesser extent, PIP_2_ [73]. Protein–protein interactions in extracellular environments can also be mediated by *SH3 domains*. As an example, the MIA protein interacts directly with extracellular matrix molecules via its SH3 domain, which comprises a new binding pocket opposite the canonical binding site, resulting in cell separation and metastasis [74].

Although it is not an exhaustive compilation, the list above comprises several extensively researched binding partners of SH3 domains. The typical proline-containing sequences recognized by these domains are part of a broader group of protein–protein interaction sites, which are well-known for their capacity to selectively bind to modular domains. The specific recognition patterns can vary depending on the specific SH3 domain and its interacting partner in canonical proline-dependent interactions. In general, the binding site of the SH3 domain is highly conserved across different SH3 domains, allowing it to bind to a variety of proline-rich sequences with high specificity. For proline-rich independent interactions on the structure of SH3 domains, binding can vary depending on the specific features of binding moieties. The binding site for proline-rich independent interactions on SH3 domains is not uniform or absolute. It is intricate and varies based on numerous factors, including the amino acid sequence, the conformation of the SH3 domain, and the target protein.

It is worth mentioning that different SH3 domains may have distinct binding specificities, and a single target protein can be recognized by several SH3 domains with different binding sites. Furthermore, the provided findings suggest that SH3 domains do not solely dictate their interaction partners. Instead, they have a complex impact on protein–protein interactions that cannot be accurately predicted based solely on their intrinsic specificity [6,75]. The specificity of SH3-dependent interactions in living cells can be determined by various factors including SH3 domain features, surrounding amino acids, a combination of multiple SH3 domains and peptide motifs, the co-expression and co-localization of SH3 domain proteins and partners, allosteric intramolecular interactions, and protein context, which includes their position within the host protein and potential intramolecular interactions [75]. This highlights the existence of intricate interactions between SH3 domains and their respective targets [6]. The interplay between the SH3 domain and the target protein is crucial for establishing specificity in protein–protein interaction networks, shedding light on how these networks evolve, and their relevance to diseases like cancer. Multi-domain analysis and the classification of human SH3CPs is essential for comprehending the patterns and characteristics of SH3 domains within a protein context, enabling a deeper comprehension of potential specificity and intramolecular interactions.

## 3. DCPs Belong to a Versatile Superfamily

Proteins containing SH3 domains are frequently identified by the similarity of their sequences [31]. From a total number of 394,887 SH3DCPs, which are present in all organisms, 1132 are reviewed; in humans, a set of 237 out of 770 proteins have been analyzed and characterized [76]. A combination of advanced searching methods with a detailed sequence comparison, using multiple sequence alignments of inputs generated by the ClustalW algorithm, was used to review the SH3DCPs and identify the accuracy of the regions annotated as SH3 domains [77]; this yielded 298 SH3 domains embedded in 221 human SH3DCPs (Appendix A). Though the basic classification of the SH3DCP superfamily is based on their ability to interact with a specific target (Table 1), they can also be classified functionally. They encompass a wide range of protein families that are highly divergent in terms of function and size, however, they are only somewhat well-characterized. Cell processes involving SH3DCPs are regulated in many types of tissues and cells. Gene Ontology describes SH3DCPs in terms of three independent categories: biological process, molecular function, and protein class (Appendix A). These categories are distributed across three compartments: cytosol, extracellular, and nucleus.

These proteins are typically located at the interface between cytosol and membranes, especially plasma membranes, and they act as molecular components for the formation and stabilization of junctional complexes and synaptic connections [78,79]. SH3DCPs are also observed in a variety of scaffolding proteins, including cytoskeletal components, such as Myosin and spectrin, to maintain and regulate stability and motility [80,81]. Moreover, SH3 domains employ liquid–liquid phase separation as a mechanism for cellular compartmentalization through interactions with PRMs to arrange the constituent components of distinct pathways for the forthcoming signal transduction [82,83,84,85]. Furthermore, the MIA protein family consists of secreted extracellular proteins that contain a single SH3 domain, with a conserved SH3 domain-like fold, supplemented by a beta paralleled beta-sheet and two disulfide bonds. These proteins serve as extracellular matrix constituents that are essential for tissue reorganization and cellular attachment [86,87].

SH3DCPs also control the molecular functions of enzymes, receptor activities, and transport processes [88,89]. SH3 domains are protein binding modules in enzymes like phospholipase Cγ [90]. Adaptor and docking SH3DCPs are involved with influencing signaling pathways, including non-receptor tyrosine kinases of the SRC family, for the regulation of its catalytic activity and/or mediating interactions [91]. It is important to mention that SH3DCP can enter the nucleus under certain circumstances. One such example is when CASK acts as a molecular regulatory coactivator of Tbr-1 to induce transcription of T-element-containing genes, such as reelin, which is required for cerebrocortical development [92]. However, the CASK-GK domain is enough for this interaction, and further research is needed to fully understand the involvement of indirect effects or interactions between other proteins with the SH3 domain in the co-activation of Tbr-1 by CASK. In another example, through the SH3 domain, SPTAN1 (αII-spectrin) could potentially contribute to the repair of DNA interstrand cross-links in the nucleus [93]. Hence, this implies that the SH3 domain serves as a mediator of complex formation, linking signaling proteins at the right time and in the right place with the corresponding signaling pathways [31]. Figure 2 depicts the formation of various protein complexes through SH3 domain-mediated protein–protein interactions, which bind to partner proteins and play a general role in different signaling pathways. Biologically, the roles of proteins from SH3DCPs are vastly diverse, ranging from signaling pathways related to proliferation, cell survival, cell growth, actin reorganization, cell migration, endocytosis, apoptosis regulation, and proteasome degradation (Figure 2A). SH3 domains mediate the involvement of numerous proteins both upstream and downstream of the EGFR-receptor tyrosine kinase (RTK). For instance, GRB2, NCK, BTK, and SRC SH3 domains interact with EGFR, resulting in the activation of downstream pathways involved in cell proliferation and actin reorganization. In addition, a huge number of SH3CPs regulate actin dynamics and cell migration via the direct mediation of SH3 domains. The role of complex formations, mediated by SH3 domains, is also clear in Vesicular trafficking. Moreover, several complexes that are mediated by SH3 also contribute to T-cell function, immune responses, muscle contraction, and synaptic activity (Figure 2B).

The number and nature of domains in many SH3DCPs are striking, especially the abundance of lipid membrane binding domains, along with protein interaction domains, such as SH2, WW, and Ig-like domains, and a large number of catalytic and regulatory domains, such as kinase, REM, GAP and GEF domains (Appendix A). In particular, ITSN1/2 (also known as EHSH1 or SH3P17, and SH3P18; Appendix A) and DNMBP (also known as ARHGEF36 or TUBA) are CDC42GEFs, and they contain five and six SH3 domains, respectively (Appendix A) [94]. They play crucial roles in linking Exo-/endocytosis, actin dynamics, and signal transduction through the small GTPase of the RHO family [95,96,97,98,99]. The association of the C-terminal SH3 domain of DNMBP (TUBA) with the N-terminal cytoplasmic PRM of tricellulin (PLPPPPLPLQPP; aa 46–57) results in TUBA-mediated CDC42 activation, which is required for the regulation of junctional tension in epithelial cells [100]. In addition, ITSN1 recruits Endophilin 1 (SH3GL2) at sites of clathrin-mediated synaptic vesicle recycling via an SH3-SH3 domain-mediated complex formation. The second SH3 domain of ITSN1 appears to be essential for endophilin1-SH3 interactions in this process [55].

The SH3DCPs are available in a wide range of molecular weights. The largest SH3DCP is OBSCN (obscurin or ARHGEF30; approximately 720 kDa), a giant sarcomeric protein of the RHOGEF family that interacts with calmodulin and titin [101]. OBSCN contains mainly I-set (Ig domains) which provide segmental flexible binding sites for proteins like titin during the assembly of the sarcomere, as well as the SH3 domain near the tandem DBL homology (DH)/RHOGEF and pleckstrin homology (PH) domains. Interestingly, a polyproline stretch within the DH domain has been proposed as a potential regulatory component as it acts like an intramolecular ligand in the SH3 domain [101]. The observation that CaMKII selectively phosphorylates the isolated SH3 domain, but not the SH3-DH fragment, suggests the presence of functional interplay between the SH3 and DH domains and their potential influence on phosphorylation events in obscurin. However, the role of the DH and SH3 domains with regard to the functioning of obscurin appears to be intricate and dependent on various factors [102]. Additionally, investigations into UNC-89, a *C. elegans* counterpart of human OBSCN, have disclosed its location at the sarcomeric M-line of the muscle. It interacts with paramyosin via the SH3 domain, and when the SH3 domain is overexpressed, it results in paramyosin mislocalization [62]. Another giant filamentous SH3DCP is NEB (Nebulin; isoform size varies from 600 to 800 kDa), which has an SH3 domain preceded by a Serine-rich region, both of which are essentially involved in the interactions between several key signaling molecules (e.g., titin, N-WASP, α-actinin, myopalladin, and zyxin). These interactions allow for the association of NEB with the sarcomeric Z-line in skeletal and cardiac muscles, and the regulation of thin filament lengths and contractility [103]. Two other giant proteins belonging to the plakin family are MACF1 (ACF7; 620 kDa) and DST (dystonin or BPAG1; 629.78 kDa), which are responsible for interacting with a variety of signaling proteins, and they provide the versatility to create links between different components of the cytoskeleton, including actin microfilaments, microtubules, and intermediate filaments [104]. An SH3 domain is positioned within the central region of the plectin domain in these proteins, and it has been proposed to interact with the SR4 domain intermolecularly to stabilize the structure and aid with ligand-binding affinities, particularly in plectin or other plectin family members such as MACF1 and DST [70]. This indicates that the SH3 domain plays a crucial role in facilitating interactions and functional interplay within these large proteins, likely contributing to their stability, ligand-binding affinity, and overall functionality. On the other hand, GRAPL, OTOR, and MIA are the smallest SH3DCPs, with molecular weights of 13.44, 14.33, and 14.5 kDa, respectively. GRAPL is similar to GRAP1/2 and GRB2 proteins, but it contains only one SH3 and one SH2 domain (Appendix A). It is likely involved with linking intracellular tyrosine kinase signals to RAS GTPases. OTOR (otoraplin or MIAL1) and MIA belong to several extracellular SH3DCPs of the melanoma-inhibiting activity (MIA) family [86], and they contain only one SH3 domain. A crucial question concerns the role of such ‘mini-proteins’ and what they are, as well as how they are involved in extracellular processes. MIA has been designated as a cartilage-derived retinoic acid-sensitive protein that is mainly secreted as an 11-kDa protein in cartilage tissue during embryogenesis and adulthood [105]. In this respect, MIA appears to influence the action of bone morphogenetic protein 2 and transforming growth factor beta 3 during mesenchymal stem cell differentiation by promoting the chondrogenic phenotype and inhibiting osteogenic differentiation [105]. MIA interacts with fibronectin during this process, and it competes with integrin binding, detaching cells from the extracellular matrix [106].

## 4. Phylogenetic Classification of SH3DCPs

The next question we addressed concerned how to classify or categorize SH3DCPs, taking into account their heterogeneous domain composition. As the phylogenetic tree based on similarities of isolated SH3 domains was not of practical use, we have used an approach based on the similarities of domain compositions between SH3DCPs. For this purpose, primary sequences of an entire collection of 221 human SH3DCPs were first retrieved from the UniProt database, and they were analyzed for occurrences of protein domains. Next, mutual similarities in terms of domain composition between all protein pairs in the collection were evaluated. The resultant matrix was then subjected to phylogenetic analysis using MEGA software (version 7.0). The final phylogenic tree shed light on the evolutionary relationships between the human SH3DCP superfamily, and it allowed the superfamily to be classified into thirteen different SH3DCP families (Figure 3). An inspection of individual families, based on the respective domain organizations (Appendix A), revealed the following findings. (i) They differ in terms of the number of SH3DCPs per family, ranging from 2 (family 6) to 54 (family 3). (ii) The classification of SH3DCPs into individual families is often based on the combination of the SH3 domain with at least one or two similar domains, for example, SH2 and/or KinYST domains (Family 1); membrane-binding BAR and PH domains, RHOGAP, or RHOGEF domains (Family 2); single or several SH3 domains combined with other shared domains (Family 3); PDZ and/or the GuaKin domain (Family 4); FCH and/or RHOGAP domains (Family 5); UBA and HPhos domains (Family 6), S-rich and CAS-C domains (Family 7); Myosin and/or MyTH4 (Family 8); SAM* along with PTB and SLY in some SH3CPs (Family 9); spectrin domain and EF-hand (Family 11); DOCK and DHR domains (Family 12); and some were also classified with only a single SH3 domain (Family 13), except Family 10, which comprises diverse combinations of the SH3 domain. (iii) Exploiting combinations of SH3, with specific domains in each family of the SH3CPs’ domain–organization, indicates that the parallel domain–combination is evolving. This also explains the functional differentiation of the SH3 domain in different pathways. (iv) *SH3 domains* can function as adaptors, scaffolds, modulators, and regulatory domains.

### 4.1. Family 1

Proteins belonging to Family 1 share a mostly conserved domain called the tyrosine kinase domain, which is responsible for their catalytic activity and phosphorylation of target proteins. They can be classified into four groups of non-receptor Tyrosine Kinases (SRC, FYN, YES, HCK, LCK, BLK, FGR, FRK, SRMS, BTK, ITK, TEC, TXK, ABL1, ABL2, MATK, CSK, LYN, PTK6, TNK2, TNK1 [107,108]), adaptor Proteins (GRB2, GRAP, GRAP2, GRAPL, CRK, CRKL, SLA, SLA2 [109,110,111,112]), tyrosine Kinase-associated Signaling Proteins (RASA1, MAP3K21, MAP3K10, MAP3K11 [113,114], and Phospholipase C, including PLCG1 and PLCG2 [115,116]). The main feature of these proteins is that they are all involved in signal transduction pathways. More specifically, when transmitting signals from the cell surface to the cytoplasm and nucleus, they can affect gene expression and various cellular processes. The SH3 domain plays a crucial mediating interaction-based and regulatory role in this family. For example, SH3 domains of adaptor proteins, such as GRB2 and CRK, bind to proline-rich motifs in other signaling proteins, allowing them to link receptor tyrosine kinases to downstream signaling pathways [40,117]. In some cases, the SH3 domain affects the catalytic activity of the kinase domain. For instance, the SH3 domain of the non-receptor tyrosine kinase, SRC, can interact with its own SH2 domain and N-terminal fragment of the kinase domain, leading to the inactivation of its kinase activity [118].

### 4.2. Family 2

The proteins listed in Family 2 are primarily involved in the regulation of Rho family GTPases, and in some cases, those of the ARF family, which are critical for regulating the actin cytoskeleton and an array of essential cellular processes; these encompass cell migration, cell division, cell adhesion, and membrane trafficking. They can be classified further into two subcategories: GTPase-activating proteins (GAPs) and guanine nucleotide exchange factors (GEFs). GAPs are negative regulators of Rho or ARF family GTPases, and they stimulate the intrinsic GTPase activity of GTPases, which leads to their inactivation. The proteins of this family are GAPs, as follows: ARHGAP10, ARHGAP26, ARHGAP42, ARHGAP12, ARHGAP27, ARHGAP9 as RHOGAPs, and ASAP1, ASAP2 are ARF GAPs [119,120,121,122]. GEFs, on the other hand, activate GTPases by promoting the exchange of GDP for GTP. The proteins in the list are Rho-GEFs, as follows: SPATA13, ARHGEF4, ARHGEF26, NGEF, ARHGEF19, ARHGEF16, ARHGEF5, ARHGEF9, ARHGEF6, ARHGEF7 [120,123,124,125,126]. TRIO, KALRN, MCF2L, VAV1, VAV2, and VAV3 are multi-domain GEFs that regulate Rho family GTPases and other signaling pathways [120,127,128]. TRIO and KALRN activate RHO GTPases, RAC1 and RHOA, and they are involved in cell migration and differentiation [129]. VAV proteins and MCF2L activate RAC1, RHOA, and CDC42, and they are involved in cell growth, differentiation, and immune responses [127,130]. SKAP1 and SKAP2 do not have a canonical guanine nucleotide exchange factor (GEF) domain. Instead, they have been shown to act as RAP1 GTPase activators through a non-canonical mechanism that involves interactions with other proteins. More specifically, SKAP1 has been shown to bind to RIAM (RAP1-interacting adapter molecule), which, in turn, recruits activated GTP-bound RAP1 by promoting the membrane translocation of RAP1 for T-cell adhesion [131,132]. SKAP2 might also interact with RIAM, and it can similarly activate RAP1. Therefore, although SKAP1 and SKAP2 do not have a canonical GEF domain, they function as GEFs for RAP1 through protein–protein interactions with RIAM. There is a possibility that SH3 domains, similarly to other enzymes, control the activity of the GEF and GAP domains through inter/intra-molecular interactions. For example, unique characteristics were observed in this KALRN (kalirin) SH3 domain, including the presence of novel binding sites for the intramolecular PxxP ligand, as well as for binding to the adaptor protein, CRK, to inhibit the GEF activity of KALRN [133].

### 4.3. Family 3

The presence of multiple SH3 domains, in most members of Family 3, may confer several advantages, including increased specificity. First, having multiple SH3 domains with different binding specificities allows proteins to interact with a larger number of partner proteins and potentially simultaneously modulate multiple signaling pathways. Second, it might lead to cooperative binding, which means that the presence of multiple SH3 domains can allow a protein to bind to multiple sites on a single partner protein, which can enhance the affinity of the interaction and potentially stabilize protein complexes. A study conducted on a SH3RF3 protein from this family used a detailed functional scaffolding analysis that revealed that its fourth SH3 domain interacts with MKK7. Additionally, it was found that the first and second SH3 domains of SH3RF3 interact with JIP3 and JNK1. These findings suggest that SH3RF3 plays an important part in aiding the assembly of the MKK–JNK complex via JIP, which leads to the activation of JNK-JUN [134]. Thirdly, the regulation of protein–protein interactions occurs when the SH3 domains in a protein can interact with each other, or with other domains within the same protein, to regulate protein–protein interactions. For example, autoinhibitory interactions of SH3 domains can block binding sites and prevent interactions until a regulatory signal is received. For example, ITSN1-L, which is a RHO-GEF, plays a crucial role in regulating both endocytosis and actin cytoskeletal rearrangements, and its SH3 domains are important for controlling its exchange activity. The SH3 domains block the binding of CDC42 to the RHO-GEF domain (or DH domain) via inter-domain interactions, which inhibits exchange activity [98]. Lastly, localization concerns the presence of multiple SH3 domains with different binding specificities, which can also allow proteins to target different subcellular compartments and interact with different sets of proteins in those locations. Interestingly, the specific order and arrangement of the SH3 domains were found to be important for maintaining the integrity of protein–protein networks in SH3CPs with multiple SH3 domains [6].

Members of this family can also function as adaptor proteins that typically contain multiple domains, and they can couple together different signaling molecules or components of cellular pathways. Many proteins of Family 3 (SH3CPs) fall into this category. For example, CD2AP (CD2-associated protein) is an adaptor protein that interacts with CD2, a transmembrane receptor protein on T cells [135], and other cytosolic proteins such as nephrin, a protein important for maintaining the integrity of the glomerular filtration barrier in the kidney [136]. The SH3 domains in CD2AP are thought to mediate protein–protein interactions with other signaling molecules and cytoskeletal components [137]. NCK1 and NCK2 (non-catalytic region of tyrosine kinase) proteins are adaptor proteins that link signaling molecules with downstream effector proteins involved in cytoskeletal regulation, membrane trafficking, and gene expression [137,138]. They contain several protein-binding domains, including SH3 domains, that enable them to simultaneously interact with multiple partners. In addition, all RIMBP proteins (RIMBP2, RIMBP3, RIMBP3B, RIMBP3C) are part of the synaptic vesicle release machinery and are involved in regulating neurotransmitter release [139]. They contain several domains that allow them to interact with other proteins involved in the synaptic vesicle cycle. Another category that some members of this family fall into is signaling proteins that act as intermediates or effectors in various signaling pathways. Some examples of MAPK8IP1 and MAPK8IP2 are as follows. MAPK8IP proteins (mitogen-activated protein kinase 8 interacting protein) are involved in the regulation of the JNK (c-Jun N-terminal kinase) signaling pathway, which is important for stress responses and apoptosis [140]. The SH3 domain in these proteins mediates protein–protein interactions with upstream and downstream components of the pathway. STAC, STAC2, and STAC3 are types of STAC protein that are involved in the regulation of calcium channels, and they play a role in skeletal muscle function. They contain several domains, including the SH3 domain, that interact with different components of the calcium channel complex [141]. OSTF1 (Osteoclast-stimulating factor 1) is another example of the protein involved in the regulation of bone resorption by osteoclasts [142]. The SH3 domain in OSTF1 is thought to mediate interactions using signaling molecules involved in the regulation of osteoclast activity. SH3 domain-containing cytoskeletal proteins, categorized as cytoskeletal proteins, such as Endophilins (Endophilin A2, Endophilin B1, Endophilin B2, Endophilin 1, Endophilin 3), are involved with controlling the organization and dynamics of the cell cytoskeleton. They are involved in the formation and recycling of clathrin-coated vesicles and the regulation of the actin cytoskeleton. Endophilins contain, among others, a BAR domain which further contributes to their membrane curvature recognition [143]. They also interact with proteins such as dynamin and synaptojanin via the SH3 domain which regulates the formation of clathrin-coated vesicles during endocytosis [144,145]. DNMBP or TUBA (Dynamin-binding protein) is also involved in actin cytoskeleton organization, and it is thought to play a role in endocytosis. The SH3 domains in DNMBP are involved in protein–protein interactions with other cytoskeletal and signaling proteins [95].

### 4.4. Family 4

Proteins listed in this family share SH3 domains and/or PDZ and/or Guanylate Kinase (GuaKin/GK) domains, and they often have similar functions associated with the regulation of protein complexes and the structure and function of the synapse, a junction between two neurons that allows for the transmission of information. SHANK1, SHANK2, and SHANK3 are scaffolding proteins that play a crucial role in the organization and function of the postsynaptic density (PSD), a protein-rich area of the synapse [146]. SHANK proteins interact with other proteins to anchor neurotransmitter receptors and signaling molecules in the PSD, thereby regulating the strength of synaptic transmission [147]. MPP1, MPP2, MPP3, MPP4, MPP7, PALS1, and PALS2 are members of the membrane-associated guanylate kinase (MAGUK) family in synapse organization and function. MAGUK proteins interact with other proteins to form a complex network of signaling molecules at the synapse, thereby regulating synaptic transmission and plasticity [148,149]. CASK is a protein in the same subfamily that interacts with other synaptic proteins, including β-neurexins, and Rabphilin3a via the PDZ domain; it plays a role in the regulation of neurotransmitter release [148]. DLG1, DLG2, DLG3, DLG4, and DLG5 are members of the Discs Large (DLG) subfamily of proteins belonging to the MAGUK family; they are involved in the formation and control of neurotransmitter release [150,151]. CACNB1, CACNB2, CACNB3, and CACNB4 are subunits of voltage-gated calcium channels (VGCCs) belonging to the MAGUK family; they regulate the entry of calcium ions into neurons. Calcium influx through VGCCs is important for synaptic plasticity and neurotransmitter release [152]. In contrast, TJP1, TJP2, and TJP3, which belong to the ZO subfamily, are also members of the MAGUK family. They are not expressed in neurons, but in the brain, and they play a crucial role in maintaining the blood–brain barrier [153].

Overall, although the SH3 domain’s interaction with its targets is less understood compared with PDZ domains, studies on MAGUK proteins, such as DLG, provide insights into the complex regulation of SH3 domain interactions and their potential roles in cellular processes. The N-terminal region of the human DLG undergoes alternative proline-rich region insertion splicing that can bind in vitro to multiple SH3 domains and control the formation of protein clusters [154]. For example, the N-terminal portion of DLG1 (SAP-97) can bind to the SH3 segment of DLG4 (PSD-95), indicating a potential heteromeric interaction between these two proteins. This interaction may play a role in dendritic clustering and the trafficking of GluR-A AMPA receptors [155]. Other studies suggest that the SH3 domain of DLG1 (SAP-97) and DLG4 (PSD-95) forms a specific interaction with its GK domain, and this intramolecular interaction prevents intermolecular associations; this sheds light on the role of the SH3 domain with regard to MAGUK function and oligomerization [156,157,158]. Recent findings suggest that the SH3 domain modulates the GK domain through an allosteric mechanism rather than by blocking the GK binding surface [159]. The SH3-HOOK-GK domain configuration is present in most MAGUK proteins, suggesting that this interaction is a shared characteristic among MAGUK proteins [157,160]. Overall, these proteins play important roles in regulating the organization and function of the synapse, and the dysregulation of their activity has been linked to various neurological and psychiatric disorders.

### 4.5. Family 5

Proteins classified into this family share a similar domain architecture. They all contain at least one SH3 domain, either a FCH or RHOGAP domain, or both. This combination of domains is unique to this protein family and sets them apart from other proteins. The combination of the SH3 domain with the FCH and/or RHOGAP domains in this group of proteins suggests that they may play a role in regulating actin cytoskeleton dynamics and membrane trafficking. Proteins containing the FCH domain may participate in protein–protein interactions, and they may potentially contribute to the organization of RHO proteins and the actin cytoskeleton [161]. Conversely, the RHOGAP domain regulates RHO family GTPases, which are important regulators of actin cytoskeleton dynamics.

### 4.6. Family 6

UBASH3A and UBASH3B are two proteins that belong to the same protein family, called the Ubiquitin-associated and SH3 domain-containing protein (UBASH3) family. These proteins are involved in the regulation of signal transduction pathways, including T-cell receptor signaling and cytokine production [162,163,164]. Functionally, both UBASH3A and UBASH3B contain an Ubiquitin-associated (UBA) domain and a SRC homology 3 (SH3) domain. The UBA domain enables the interaction between these proteins and ubiquitin, a protein that plays a critical role in the regulation of protein degradation, DNA repair, and immune response [165]. The SH3 domain allows UBASH3A and UBASH3B to bind to proline-rich motifs in other proteins, including signaling proteins, receptors, and enzymes, thereby regulating their activity [163,166,167]. Structurally, UBASH3A and UBASH3B are similar in size, each consisting of 504 amino acid residues. Both proteins share a high degree of sequence identity, with 80% sequence similarity. The overall structure of these proteins is similar, with an N-terminal UBA domain followed by a central SH3 domain and a C-terminal HPhos region. However, there are some differences in the sequence and structure of the UBA and SH3 domains between UBASH3A and UBASH3B, which may contribute to their distinct functions.

### 4.7. Family 7

BCAR1, NEDD9, and CASS4 also share other domains in addition to the SH3 domain, namely, the S-RICH and CAS-C domains. The S-RICH domain, which is a stretch of amino acids enriched with serine residues, is located in the N-terminal region of all three proteins. It has been shown to be crucial for the localization and activity of these proteins at focal adhesions, which are sites of cell adhesion and signaling, by binding to 14-3-3 proteins [168]. The CAS-C domain is a domain that is found in the C-terminal region of all three proteins, and it assists with binding to other signaling molecules, such as the adapter protein, SRC, which mediates downstream signaling events [169,170]. Therefore, the common structural feature of the SH3 domain, coupled with the S-RICH and CAS-C domains, contributes to the functional similarities between BCAR1, NEDD9, and CASS4; this is because it allows them to interact with the other proteins involved in cell adhesion and signaling pathways, leading to similar functional roles in terms of regulating cell adhesion, migration, and proliferation [169].

### 4.8. Family 8

These proteins share common structural features in that they all contain SH3 with myosin domains belonging to the myosin superfamily. Myosins are a family of motor proteins that use the energy from ATP hydrolysis to generate force and move along actin filaments, resulting in the generation of force and motion [171]. Functionally, myosins are involved in a wide range of cellular processes, including muscle contraction, cell migration, membrane trafficking, and organelle transport [172]. The specific functions of listed myosins may vary depending on their expression patterns, subcellular localization, and interactions with other proteins. For example, MYO7A is involved in hearing and balance [173], whereas MYO5A is involved in melanosome transport and pigmentation [174]. Other myosins, such as MYO1E, are involved in cell migration and the regulation of the actin cytoskeleton [175]. Myosins can be divided into two broad categories, as follows: conventional and unconventional myosins. Conventional myosins are typically found in muscle tissue and are responsible for generating the force and movement required for muscle contraction [176]. Unconventional myosins, on the other hand, have a more diverse range of functions, and they are found in a variety of cell types and tissues throughout the body [177]. Some examples of these unconventional roles include acting as tension sensors and dynamic tethers, organizing F-actin during endo- and exocytosis, and maintaining the mitotic spindle structure [178]. Unconventional myosins often have a more complex domain structure than conventional myosins, and the SH3 domain is one of the additional domains that is commonly found in these proteins [177]. The SH3 domain in some myosins, given their interaction with other proteins, may carry out these functions [80]. All listed myosins contain a single SH3 domain, which is involved in mediating protein–protein interactions, and this is consistent with the idea that these myosins play roles in diverse unconventional processes. For example, MYO1E, which is an unconventional myosin involved in the cell migration and regulation of the actin cytoskeleton, contains a SH3 domain that has been shown to interact with a protein called ZO-1 [175]. This interaction is thought to play a role in regulating junctional integrity in kidney podocytes by contributing to the slit diaphragm complex [179]. Similarly, MYO7A, which is involved in hearing and balance, contains an SH3 domain that contributes to the interaction with the protein harmonin. This interaction is important for the localization of MYO7A to the stereocilia in the inner ear, where it is involved in generating mechanical force and movement [180].

### 4.9. Family 9

The concurrent presence of SH3 domains and SAM*, PTB, and SLY domains in some of the proteins listed in Family 9 suggests that they play roles in various aspects of signal transduction and protein–protein interactions. The SAM* (sterile alpha motif) domain is a conserved protein domain of around 70 amino acids that is present in many proteins involved in signal transduction and transcriptional regulation [181]. SAM* domains are known to mediate protein–protein interactions and are believed to function as regulatory domains that can influence the activity or localization of their associated proteins [182]. The PTB (phosphotyrosine binding) domain is another protein domain that is commonly found in signaling proteins. PTB domains bind to specific phosphorylated tyrosine residues in other proteins, and they are involved in mediating protein–protein interactions that are essential for the proper functioning of signaling pathways [183]. The SLY domain, a conserved family of lymphocyte signaling adapter proteins domain, is present in eukaryotes and is associated with SH3 and SAM domains. It is identified in various proteins, including SLY1/SASH1, SASH3, and SAMSN1 [184]. The combined presence of these domains in listed proteins suggests that they likely function as adaptors or scaffold proteins that help to assemble and organize signaling complexes, and that they mediate the protein–protein interactions that are critical for signaling and regulation. Adaptor proteins contain protein–protein interaction domains that link receptors to downstream signaling components, whereas scaffold proteins provide a physical platform for multiple signaling components to interact with and regulate each other’s activity. Based on their known functions and structural features, EPS8, EPS8L1-3 [185], SASH1 [186], SASH3, and SAMSN1 [187] are believed to function as adaptor proteins, whereas CASKIN1 and CASKIN2 are scaffold proteins. CASKIN1 and CASKIN2 contain multiple domains which enable them to function as scaffold proteins that can organize multi-protein complexes [188]. The structural investigation of CASKIN2′s SH3 domain using NMR revealed that its peptide-binding cleft differed from the typical binding sites for polyproline ligands due to the presence of non-canonical basic amino acids. Mutations in the cleft suggested that the SH3 domain in CASKIN2 may have lost its functional ability to promote protein–protein interactions beyond the conventional roles typically associated with SH3 domains [189].

### 4.10. Family 10

Although SH3 domains may be a shared feature among these proteins, their overall domain architectures and functions are diverse. Therefore, it is important to note that some of these proteins may have multiple functions, or they may interact with multiple signaling pathways; their precise classification can depend on context and experimental findings. However, they can be primarily classified into the following functional categories: signal transduction (STAM, STAM2 [190], NCKIPSD [191], MAP3K9 [192], MACC1 [193], PRMT2 [194], AHI1 [195], LASP1 [196], SGSM3 [197]), cytoskeletal remodeling (HCLS1 [198], CTTN [199], NEBL, NEB [103], LASP1 [196], FYB [200]), endocytosis (SH3TC1, SH3TC2 [201], SNX9, SNX33, SNX18 [202]), and immune system function (NCF1, NCF1B, NCF1C, NCF2, NCF4 [203], NOXO1, NOXA1 [204]). Furthermore, many of these proteins have multiple SH3 domains, and some may have other protein–protein interaction domains or motifs that contribute to their functions.

### 4.11. Family 11

The shared structural and functional features of these proteins are primarily related to their roles in cytoskeletal organization and cell adhesion. The spectrin domain is a key structural component that provides mechanical stability to the cytoskeleton. It forms a long, flexible rod-like structure that can interact with other proteins, cytoskeletal elements, and lipids to provide support and resistance against deformation [205,206,207]. The SH3 domain, on the other hand, plays a key role in cytoskeletal organization and cell adhesion by regulating protein–protein interactions and localization. The EF-hands have a high affinity for Ca2+, they undergo a conformational change when bound to it, and they are essential for maintaining the structural integrity of the skeleton [206]. Together, the SH3, spectrin, and EF-hand domains found in these proteins can work together to regulate critical protein–protein interactions that maintain the structural integrity of the cytoskeleton and regulate cellular adhesion and signaling. Although each of these proteins have unique features and functions, they all share common structural and functional elements that reflect their common ancestry and evolutionary history.

### 4.12. Family 12

These proteins share both structural and functional similarities as they all belong to the same family of guanine nucleotide exchange factors (GEFs), known as the DOCK family. Structurally, they all contain a conserved DHR-2-C (DOCK homology region 2) domain which is responsible for the GEF activity of these proteins, as well as other domains such as DHR-2-A (lipid-binding DOCK homology region) and the SH3 domain. Functionally, they play important roles in the regulation of cytoskeletal dynamics, cell migration, and immune and neural cell function [208,209]. In the DOCK family, the SH3 domain plays a regulatory role by mediating interactions with proline-rich motifs in other proteins, allowing DOCK proteins to bind to, and regulate the activity of, a variety of cytoskeletal and signaling proteins. There are some examples of how the SH3 domain in DOCK proteins can play a role in regulating protein–protein interactions. The SH3 domain of DOCK2 interacts with the PRM of ELMO1, which may relieve their autoinhibition to promote the activation of RAC in lymphocyte chemotaxis [210,211]. Moreover, the DOCK1–ELMO1 interaction was identified for the localization and regulation of RAC1 in cytoskeletal organization and cell migration [211,212]. The C-terminal PRM region of DOCK1 can also interact with the SH3 domain of several proteins, including the adaptor protein, NCKβ, and CRK, which helps to control cell migration [213,214,215]. Thus, the SH3 domain is an important structural component that facilitates these interactions to influence the subcellular activity of DOCK proteins, as well as their ability to activate downstream signaling pathways.

### 4.13. Family 13

All of these proteins contain only one SH3 domain. The specific function of each protein may be different, but they all share the ability to interact with other proteins via their SH3 domain. For example, FYB2 (FYN binding protein 2) regulates T-cell receptor signaling and is involved in the formation of the immunological synapse [216]. Another study found that MIA, a protein secreted from malignant melanoma cells, enhances melanoma cell migration and invasion by interacting with extracellular matrix proteins and integrin [87,217]. In addition, cadherin-7 was identified as a new MIA-binding protein that negatively regulates the expression and activity of MIA, and it plays a role in the migration of melanoma cells during tumor development [218]. Another review integrates research on *Drosophila Tango1* and human MIA/cTAGE proteins to provide an evolutionary perspective on ER-Golgi transport, which highlights the role of the MIA protein involved in the regulation of the ER-Golgi transport of proteins [219]. OTOR (melanoma inhibitory activity-like (alias MIAL)) may play a role in the development and maintenance of the inner ear [220]. NPHP1 (Nephrocystin-1) plays a role in the macromolecular complex formation and function of cilia, and disruptions to these complexes can cause renal cystogenesis [221]. PRAM (PML-RAR alpha-regulated adapter molecule) is involved in the regulation of the differentiation of hematopoietic cells [222]. SH3D21′s (SH3 domain-containing protein 21) function is currently unknown, and further research is needed to fully understand the specific role and mechanisms of SH3D21 with regard to signaling processes. The cellular localization of these proteins may vary depending on their specific function and the cell type in which they are expressed. Although some proteins may have a predominant localization to a particular subcellular compartment, others may be distributed more broadly throughout the cell.

## 5. SH3 Domain-Specific Disorders, Diseases, and Potential as Drug Targets

The mutational disruption of SH3-target interactions is associated with a variety of human diseases (Table 2). SH3 domain mutations have been linked to the development of various diseases such as Joubert syndrome, leukemia, lymphomas, Usher Syndrome or nonsyndromic deafness, centronuclear myopathy, schizophrenia, and other neurodevelopmental disorders (Table 2). Cancer cells can invade by inducing epithelial-to-mesenchymal transition via SH3DCPs such as SRC family kinases [91]. Elevated levels of other SH3DCPs, such as GBR2, CRK, and SAMSN1 adaptor proteins are also detectable in a large percentage of breast cancers and human colon and lung cancer samples, respectively [223,224]. In multiple in vitro experiments, SH3 domains have also been shown to be prone to amyloid fiber formation under acidic conditions, and they underwent conformational changes during the aggregation process [225,226,227]. Various identified mutations in the SH3-binding motifs can affect the function and interactions of the protein; this shows the importance of SH3 mediating interactions. For example, a novel homozygous mutation (p.Ser236Phe) in the SH3 binding motif of the STAMBP gene was found in a two-year-old boy with microcephaly-capillary malformation syndrome, leading to protein instability and the prevention of STAM binding [228]. Moreover, viral and bacterial pathogens adapt SH3 protein modules or PRMs from the host to mimic and modulate host cell signaling for their own purposes [229,230]. Interesting results have also been obtained regarding the presence of PRM located at the N-terminus of the Nef protein in HIV, which is essential for the induction and progression of AIDS-like diseases [231,232,233]. Several approaches screened host–viral relationships by identifying the potential interactions between SH3DCPs, including GRB2, FYN, NCK1, HCK, and ARHGEF7, and viral proline-rich sequences [234]. The prevalence and critical regulatory roles of SH3-PRP interactions in human diseases, coupled with the impact of SH3 domain mutational dysfunction on signaling pathways and human disease and pathogenicity, allows the exploitation of their protein–protein interaction to be a potential candidate for a new drug design [80,223,230,235,236,237]. SH3 domains can be found in oncoproteins as well as in proteins that are excessively expressed in irregular signaling pathways in cancerous cells. There may be potential for pharmacological intervention in signaling cascades to inhibit proliferation; this could occur by targeting SH3 domains, with small peptides and molecules mimicking binding, and a high degree of specificity and affinity to specific SH3 domains. These molecules may represent new cytostatic agents for proliferative diseases, but they may have difficulty distinguishing between normal and cancerous cells, and they may need to be carefully dosed to avoid completely inhibiting normal cellular growth responses [8]. In addition, although SH3 domains can recognize ligands due to their modest affinity, they exhibit limited selectivity within the SH3 family, and thus, using non-specific SH3 inhibitors may lead to the de/activation of alternative pathways and resistance. The structure of SH3 domains provides important clues for designing effective antagonists. The SH3 domain can obtain selectivity through the involvement of other regions that are not accessible for ligand interactions, thereby expanding the binding site, and in some cases, via unique SH3-ligand interactions, both of which seem strategic for future drug designs [238]. On the other hand, research indicated that the presence of SH3 domains plays a crucial role in enabling the SLAP’s ability to oppose the oncogenic activity of SRC in fibroblasts [239,240]. Achieving anti-oncogenic activity through a mechanism involving SH3 indicates a potential anti-tumor function for SH3 when counteracting oncogenic activity, in addition to its role in oncogene tumor-driven SH3CPs.

One study focused on the design of spirolactam-based peptidomimetics aimed at the SH3 domain of a LYN that produced ligands with extended conformations; this resulted in comparable binding affinities to reference peptides (XPpX motif) [295]. Moreover, scientists developed the mirror-image phage display method to identify D-peptide ligands that are enzyme-resistant. This method involved creating a mirror image version of the protein and selecting peptide molecules from a peptide library that could bind to it in a solvent (water) that does not require chiral cofactors. This method can be used to identify molecules that can bind to specific target proteins, including cyclic D-peptides, that partially obstruct the binding site of the c-SRC protein. [296]. In another study, highly selective and efficient peptides that bind to the SH3 domains of CRK and CRKL proteins were developed and tested for their ability to interfere with SH3 binding in living cells [297].

Furthermore, various laboratories have conducted sophisticated experiments using combinatorial chemistry to discover novel non-peptide ligands for SH3 domains [298]. By designing ligands that complement the topography of the binding pocket, researchers were able to discover ligands with greater selectivity and affinity for the SRC-SH3, and they also discovered specific ligands for HCK-SH3 [299,300]. By adopting a similar approach, a ligand that was designed to be an SH3 inhibitor, with a high affinity for the GRB2 SH3 domain, was obtained by replacing key prolines with non-natural N-substituted residues during ligand screening [301]. Extracellular SH3CP MIAs interact with other proteins in the extracellular matrix, particularly fibronectin (FN), to facilitate the detachment of cancer cells and promote their migration and invasion into surrounding tissues [74]. A small molecule that was discovered using a binding site prediction approach and in vitro fragment screening can disrupt the MIA–FN interaction by binding to a specific pocket on the MIA protein; it can serve as a potential target during future drug development against melanoma [74]. Moreover, 2-aminoquinolines and related compounds have been identified as potential high-affinity small molecule ligands for the SRC Homology 3 (SH3) domains; this could be useful for developing novel therapeutics that treat human diseases caused by abnormal cell signaling pathways [302]. In conclusion, the collective findings from the above studies provide insights into the application of combinatorial libraries and structural biology when elucidating the intricacies of protein–ligand interactions and the potential use of small molecule ligands as drugs.

Currently, there are no approved drugs that directly target SH3 domains. However, there is ongoing research to develop small molecule ligands that can selectively bind to SH3 domains and potentially be used as therapeutics for diseases caused by abnormal signaling pathways. The identification of compounds as potential high-affinity ligands for particular SH3 domains is a basic step toward developing such drugs. However, further testing may find their efficacy and safety in vivo unsatisfactory. To address this issue, it may be beneficial to develop new strategies that can specifically target certain interactions within one or several SH3 domains of particular SH3DCPs while avoiding cross-interactions with other SH3DCPs. This could help to minimize any unintended effects on other targets. Furthermore, the aforementioned unconventional SH3 targets present exciting opportunities for potential drug development.

Moreover, understanding the intricate details of protein–protein interactions, as exemplified by the multifaceted behavior of SH3 domains, unveils novel opportunities for therapeutic interventions. Recent research has shed light on the role of SH3 domains in mediating and regulating protein–protein interactions through their proline-rich binding grooves, as well as their opposite binding sites, characteristics that might be common among many SH3 domains. For instance, the analysis of ITSN1′s structure reveals that its SH3(E) domain exhibits two distinct binding surfaces, as follows: one interacts with the catalytic DH domain to modulate GEF activity [98], and the other specifically binds proteins containing polyproline residues to facilitate the cellular specific targeting of dynamin to endocytic complexes [56,303]. Notably, the proline binding pocket of the SH3 domain does not interfere with the inhibitory function of the SH3 domains with regard to nucleotide exchange [98]. In another example, the proline-rich region of the N-WASP has been identified as an activator of ITSN1 via its interaction with the ITSN1 SH3 domain [304]. However, this activation is not observed with recombinant ITSN1 fragments alone, suggesting the involvement of an unidentified additional protein interaction on the ITSN1 SH3 domain [98], potentially occurring on the other binding surface. These findings emphasize the potential value of both the front and back sides of SH3 domains, and both surfaces can be used as promising targets for future drug development. Additionally, the complex interplay and lack of sole dependency on intrinsic binding specificities make it difficult to design drugs that effectively target and inhibit SH3 domains. Multiple factors, including the identity of the host protein and the position of the SH3 domains, play crucial roles in determining the specificity of these interactions [6,75]. Therefore, achieving the selective inhibition of SH3 domains requires a comprehensive understanding of these factors and their intricate relationships.

## 6. Concluding Remarks

The fact that SH3 domains regulate a wide range of cellular functions raises the question regarding the specificity of SH3 domain interaction networks (Figure 2). Multiple studies noted that the interaction between SH3 domains, with canonical and non-canonical target sequences in binding partners, leads to specificity among the pool of ligands (Table 1). Remarkably, in canonical binding, proline is the only N-substituted amino acid found in nature that can form the polyproline type II (PPII) helix conformation, which exposes a binding pocket for SH3 domain residues, mainly from the RT and n-SRC loops [19,61,301]. Previous studies also revealed that the poly-proline amino acid stretch is involved in SH3 domain ligand recognition [305]. Despite intensive research, the specificity of the interaction of SH3 domains for proline-rich motifs remains unknown. Understanding the molecular basis for the specific and diverse binding of SH3 domains to PRMs will provide insights into the regulation of signaling pathways. Multiple studies have been conducted to investigate and classify the interactions between SH3 domains and various ligands, resulting in diverse categorizations based on different criteria. Cesareni and coworkers have investigated the interaction landscape of the human SH3 protein family using a combination of information extraction strategy and experimental approaches, including a type of new peptide chip technology; this occurred in order to characterize the specificity and promiscuity of proline-rich binding domains and to map their interaction network. Two main groups of SH3 domains were identified based on their interaction with similar peptide ligands, as follows: SH3 domains that bind to “classical” PxXP core motifs along with positively charged amino acids, and atypical SH3 domains that lack the core motif [23]. Sidhu also performed versatile canonical and non-canonical specificity profiling of SH3 domains using peptide-phage displays with deep sequencing in 2017 [10]. Moreover, a comprehensive analysis of SH3 domain interactions concerning the evolution of four yeast species, *Saccharomyces cerevisiae*, *Ashbya gossypii*, *Candida albicans*, and *Schizosaccharomyces*, revealed that nearly 75 percent of SH3 families generated within the phylogenetic tree have a conserved SH3 specificity profile over 400 million years of evolution [306]. Moreover, numerous SH3 domains exhibit an extended repertoire of binding sequences, known as proline-independent binding. This enables SH3DCPs to mediate a broader array of interactions, including interactions with other domains, like GAP, kinase–catalytic, basic rich (BR), Guanylate Kinase (GuaKin/GK), SH3, DH, SH2, PX, and LIM4, or other targets like RNA, helixes, arginine–lysine residues, spectrin repeat, lipid, and extracellular matrix molecules. It is important to highlight that these non-traditional targets may hold substantial promise as viable candidates regarding future drug development. In recent years, to better understand the mechanisms underlying SH3-mediated cellular responses, numerous attempts to develop different methodologies for studying and mapping SH3-PRM dependent and independent binding have been conducted. Nevertheless, it can still be argued that the function of most proteins is intimately dependent upon their native tertiary structures [307]. The systematic analysis of the sequence–structure–function relationships of SH3-PRM interactions, coupled with biochemical annotations, is needed to explore correct functional sites and categories from a structure-based perspective.

Our work illustrates the evolutionary relationship of the 221 human SH3DCP superfamily, and it allows for the functional classification of these proteins into thirteen families. Such classifications provide insights into their diverse roles and interactions within cellular processes. Furthermore, it allows us to identify patterns of SH3 domains and their co-occurrence with other domains in multidomain proteins, and it allows us to uncover potential functional modules or regulatory units within proteins. This classification approach aids in the understanding of SH3 domain-mediated interactions and their contributions to intramolecular activation and deactivation, intermolecular inhibition or networking, as well as their role as scaffolding and adaptor elements in cellular function and disease mechanisms. Moreover, the potential of targeting SH3 domains, for future drug designs, as presented in this review, will help to develop novel therapeutic approaches. Several in vitro strategies for designing peptide and non-peptide targets, such as peptidomimetics, mirror image phage display, and combinatorial chemistry, have been explored in order to design ligands with enhanced affinity and selectivity for specific SH3 domains regarding the inhibition of their protein–protein interaction. Challenges such as selectivity and specificity need to be addressed when designing inhibitors, as non-specific inhibition may lead to the deactivation of alternative pathways and resistance. Moreover, uncovering the full potential of non-canonical SH3 domain binding targets may provide new possibilities for therapeutic interventions. Furthermore, the intricate interplay between, and absence of an exclusive reliance upon intrinsic binding specificities pose challenges for the development of drugs that can precisely target and inhibit SH3 domains. Continued research and exploration into SH3 domain interactions hold great promise for the future of treating diseases caused by abnormal signaling pathways.

## Figures and Tables

**Figure 1 cells-12-02054-f001:**
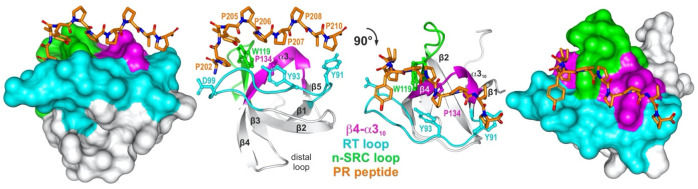
A representative structure of an SH3 domain PRM complex. A detailed view into the structure (PDB code: 1FYN) of the SH3 domain of FYN tyrosine kinase (left: surface representation; right: ribbon representation; UniProt ID: P06241) in complex with 3BP-2 PR peptide (PAYPPPPVP; orange; UniProt ID: P78314) which shows the characteristic arrangement of beta strands and the PRM-interacting variable loops, referred to as β4-α3_10_ (magenta), RT (cyan), and hydrophobic patch (W1190) flanked by n-SRC loop (green). Conserved residues that are crucial for the interaction are Y91, Y93, D99, W119, and P134. FYN SH3 shows the typical topology of two perpendicular three-stranded β-sheets and a single turn of α3_10_.

**Figure 2 cells-12-02054-f002:**
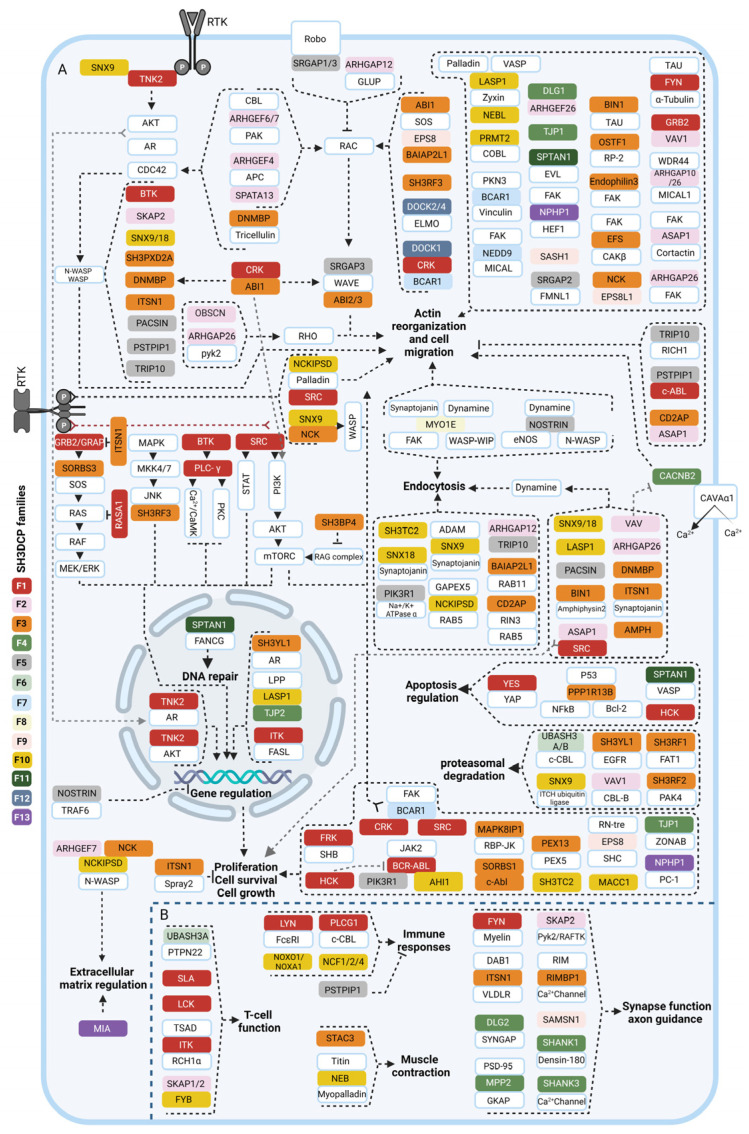
Schematic diagram of SH3DCPs in diverse signaling pathways. SH3DCPs are crucial signaling proteins, and they include adaptor proteins, kinases, RAS GEFs, RAS GAPs, scaffold proteins, and effectors. They are involved in various signaling processes in the cell. The dashed separation in the figure distinguishes general cellular functions (**A**) from specific functions in various cell types (**B**). All information presented in this figure is cited as references in Appendix A.

**Figure 3 cells-12-02054-f003:**
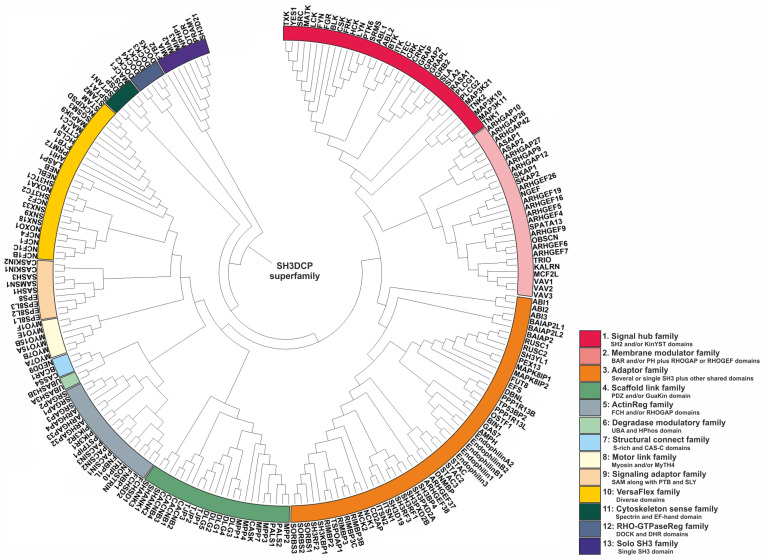
Phylogenetic tree of the SH3DCP superfamily. The tree was generated on the basis of the similarities between domain compositions. The SH3DCP superfamily can be divided into thirteen families, which are marked by different colors of classes at the outer ring.

**Table 1 cells-12-02054-t001:** Binding Specificity of SH3 domains.

Binding	Class	Ligand	SH3DP Example	Ref.
**Canonical**	Class I	+xXPxXP	MYO1E interaction with FAK-PRM1	[41]
	Class II	XPxXPx+	CD2AP-2nd-SH3 in interaction with RIN3	[42]
	Class I/II	Specificity of both ligands of I/II	FYN interaction with different PRMs Tau	[43]
	Class III	Combination of proline with non-proline residues	GRAP2-SH3C (MONA) and GRB2-SH3N interaction with HPK1 STAM2 interaction with UBPY; GRB2-SH3C interaction with SLP-7; EPS8 interaction with ABI1 (E3B1) and RN-tre; NCK1/2 N-Terminal-SH3 interaction with cytoplasmic tail of CD3ε	[46,47,48,49,50,51]
**Non-canonical**	Class VI	GAP domain	RASA1 interaction with DLC1-GAP domain	[52]
		Kinase–Catalytic domain	RASA1 interaction with Aurora kinases-catalytic domain	[53]
		Guanylate–Kinase domain (GuaKin/GK)	DLG4 (PSD-95) inter-domain interaction	[54]
		SH3 domain	ITSN1 and SH3GL2 SH3-SH3 domain complex	[55]
		DH domain	ITSN1 interaction with internal domain	[56]
		RNA	ITSN1 SH3D interaction with mRNA	[57]
		Helix	NCF2 (p67phox) SH3D interaction with NCF1(p47phox) N-term helical region; PEX13 interaction with helical segment of Pex5p; UNC-89 SH3 (homologs of human OBSCN-SH3) interaction with Paramyosin (homologs of human MYH7-skip2) coiled α-helical structures	[58,59,62,65]
		Arginine–Lysine residues	FYN-SH3 interaction with SKAP55; Gads-SH3 and GRB2-SH3_C-term_ interaction with SLP-76; BIN1-SH3 interaction with internal domain; ITSN1 interaction with CdGAP-Basic rich domain (xKx(K/R)K motif)	[28,48,63,64,65,66]
		SH2 domain	FYN-SH3 interaction with SAP-SH2	[67,68]
		PX domain	NCF1(p47phox) inter-domain interaction	[69]
		Spectrin repeat	MACF1 inter-domain interaction	[70]
		LIM4 domain	NCK-2 SH3.3 interaction with PINCH-1	[71]
		Lipid	CASKIN1 interaction with lysophosphatidic acid (LPA) PRAM1-SH3 interaction with PI(4)P and PIP2	[72,73]
		Extracellular matrix molecules	MIA-SH3	[74]

**Table 2 cells-12-02054-t002:** Diseases associated with the SH3 family.

SH3DCP	Mutation	Disease	Refs.
AHI1	FsX1103	Joubert syndrome	[241]
	Deletion of the SH3 domain	Leukemia and lymphomas	[242]
MYO7A	Missense mutation (A1628S) and truncation/deletion mutations (c.4838delA, c.5146-5148delGAG)	Usher Syndrome or nonsyndromic deafness (DFNB2)	[243,244]
FRK	R64Q	Cervix and vulva cancer	[245]
YES1	K113Q	Breast and colon cancer	[245]
ACK1	M393T, M409I	Colon, Gastric adenocarcinoma	[245,246,247,248]
AMPH	Q434X, K436X, Q573X, K575X	Centronuclear Myopathy	[249,250]
ARHGAP10	Lacking the RHOGAP and SH3 domains	Schizophrenia	[251]
ARHGEF9	G55A	Hzperekplexia, seizures or epilepsy, developmental Delay, or intellectual disability	[252]
ARHGEF23	Missense and nonsense mutations	Neurodevelopmental disorders	[253]
ARHGEF30	V5668A	Breast cancer	[101]
	A5660V	Cardiomyopathies	[254]
CD2AP	K301M	Sporadic nephrotic syndrome and focal segmental glomerulosclerosis (FSGS)	[255]
BIN1	Q434X, K436X, Q573X, K575X, P593HfsX54, X594DfsX53	Centronuclear myopathy (CNM)	[249,256]
	rs138047593 (K358 R (KR))	Alzheimer’s disease	[257,258]
BLK	A71T	Autoimmune diseases, (e.g., systemic lupus erythematosus (SLE))	[259]
BTK	Deletion of C-terminal 14 aa residues of SH3 domain	X-linked agammaglobulinemia (XLA)	[260,261]
LYN	SH3 mutations (transformative and non-transformative)	Cancer (uterine, sarcoma, thyroid, liver, head and neck, melanoma, lung, glioma, kidney, breast, hematologic)	[39]
MIA	High expression	Melanoma development, progression and metastasis	[74]
MYO15A	G2909S, G2941Vfs*94, W2931Gfs*103, R2923*, P2880Rfs*19, R2903*, R2924H, G2938R, V2940fs*3034	Human Deafness	[262,263]
NPHP1	2q13	Autosomal recessive cystic kidney disease	[264]
	L180P	Familial Juvenile Nephronophthisis	[265]
PEX13	Missense mutation at SH3, nonsense mutation of W234ter, temperature sensitive mutation of I326T, W313G	Peroxisome-biogenesis disorders (PBDs) including Zellweger syndrome (ZS), neonatal adrenoleukodystrophy (NALD), and infantile Refsum disease	[266,267,268,269]
PLC-γ1	Substitution mutations	Adult T cell leukemia/lymphoma, angioimmunoblastic T-cell lymphomas, T-cell prolymphocytic leukemia, Sézary Syndrome, PLAID, autoinflammation, immune deficiency	[270,271,272,273]
PSTPIP1	D384G, G403E, G403R, R405C	Autoinflammatory diseases (most notably in the PAPA syndrome; pyogenic sterile arthritis, pyoderma gangrenosum, and acne) and CVID (common variable immunodeficiency)	[274,275]
PTK6	L16F	Cancer	[276]
RASA1	Missense, nonsense, frame shift, and splice site mutation	Cancer, capillary malformation (CM)	[277,278]
RIMBP1	G1808S	Autosomal recessive dystonia	[279]
SASH1	S587R, M595T, E617K, I586M, S587R, M595T	DUH (dyschromatosis universalis hereditaria) and lentiginous phenotype	[280,281]
SH3PXD2B	Non-synonymous coding sequence variations (G245R, E396K, G481R)	Axenfeld–Rieger syndrome	[282]
	BDCS3 deletion (deletion of two C-terminus SH3 domains)	Borrone dermato-cardio-skeletal syndrome	[283]
SHANK1	R874H	Autism spectrum disorder	[284]
SHANK2	S557N, R569H	Autism spectrum disorder	[285]
SHANK3	Lacking parts of the SH3 domain in case of G1527A	Autism spectrum disorder and intellectual disability (ID)	[286]
SPTAN1	D2303_L2305dup	Epileptic encephalopathy	[287]
VAV1	L801P	Cancer	[288]
STAC3	W > S substitution	Native American myopathy	[289]
	P269R, N281S, W284S, F295L, H311R, K329N	Native American myopathy, dystrophin-deficient muscles	[289,290,291,292]
OBSCN	V5668A	Breast cancer	[293]
CASK	G659D	Severe intellectual disability (ID), microcephaly and pontine, and cerebellar hypoplasia in girls (MICPCH)	[294]

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
