# Peer review of "A Systematic Compilation of Human SH3 Domains: A Versatile Superfamily in Cellular Signaling"

_cells, 2023, doi:10.3390/cells12162054_

Round 1

Reviewer 1 Report

The Authors made a tremendous effort to prepare this very interesting overview about SH3 domain. If anyone is interested in starting research about how diverse SH3 domain is I would recommend this article. Supporting information is so densely packed with information that it could be mini-review on its own. However, there are minor points which should be corrected in manuscript:

1. Citation style! Citations should be in uniform citation style and there should NOT be repetition of citation. For example 26 and 27 is repetition.  Please check ALL citation for double occurrence. The journals are either in abbreviated form or full, please unify.  

2. Authors use term alpha310, I think they mean helix-3 10. Please correct.

3. Sentence: Proline is non-essential amino acid with ... Non-essential for what?

4. On Page 3 line 97 +xXPxXP and XPxXPx+ is the "+" Arg/Lys or "+x/x+" Please explain to avoid confusion.

5. Abbreviation SAM can be interpreted in two ways (as even Authors mention it in Supplementary materials). Please distinguish them in article which protein you mean. Maybe denote one with ' or *?

6. In Figure 2 Family 6 is missing please add it (panel B). Why complex DOCK1/CRK/BCAR1 is mentioned twice?

7. In Supporting information please add title and Authors. Supplementary information for " title", Authors.

8. Please unify citation style in Supplementary Information.

Author Response

The Authors made a tremendous effort to prepare this very interesting overview about SH3 domain. If anyone is interested in starting research about how diverse SH3 domain is I would recommend this article. Supporting information is so densely packed with information that it could be mini-review on its own. However, there are minor points which should be corrected in manuscript:

Authors' Response: We express our sincere gratitude for your invaluable expert opinion and constructive evaluation of our manuscript. Your feedback has significantly enhanced the overall quality of the paper. The revised version of the manuscript incorporates the changes, highlighted in yellow for your convenience. We have also included the revised text in this letter (marked in yellow) for easy reference.

1. Citation style! Citations should be in uniform citation style and there should NOT be repetition of citation. For example, 26 and 27 is repetition.  Please check ALL citation for double occurrence. The journals are either in abbreviated form or full, please unify.

Authors’ response: Thank you for bringing the citation style issue to our attention. We have thoroughly reviewed the citation and ensured that there are no duplications across all citations. To maintain uniformity, we have opted for the abbreviation format of journal names across all citation.

2. Authors use term alpha310, I think they mean helix-3 10. Please correct.

Authors’ response: Thank you for your comment. We have now addressed and revised this issue. In line 29, the term alpa310 has been revised to 310-helix to accurately refer to helix 310.

3. Sentence: Proline is non-essential amino acid with ... Non-essential for what?

Authors’ response: Thank you for bringing this to our attention. The phrase "proline is a non-essential amino acid" refers to the ability of the cells in our bodies to synthesize proline themselves, which is actually redundant to this article and has therefore been omitted from the text. We have changed the text as follows (line 61): PRMs are typically composed of proline (P) and hydrophobic (X) amino acids, with a core canonical motif XPxXP (where x can be any amino acid). The distinctive cyclic structure of proline's side chain gives proline an exceptional conformational rigidity compared to other amino acids. This unique structural property of proline may interfere with the regular formation of secondary structures, making it more abundant in unstructured regions.

4. On Page 3 line 97 +xXPxXP and XPxXPx+ is the "+" Arg/Lys or "+x/x+" Please explain to avoid confusion.

Authors’ response: Thank you for your suggestion. We have now provided an explanation and elaborated on the notation +x/x+ as follows (line 98): The location of this basic residue designated as +x/x+ determines the orientation of peptide binding in relation to the conserved proline residues at the N-terminal (+xXPxXP, class I) or the C-terminal (XPxXPx+, class II) positions of PxXP core 26,34,35.

5. Abbreviation SAM can be interpreted in two ways (as even Authors mention it in Supplementary materials). Please distinguish them in article which protein you mean. Maybe denote one with ' or *?

Authors’ response: Thank you for highlighting this point. We now use "SAM*" (page 16) to refer specifically to the field in the article and to distinguish it from the broader interpretation of SAM protein.

6. In Figure 2 Family 6 is missing please add it (panel B). Why complex DOCK1/CRK/BCAR1 is mentioned twice?

Authors’ response: Thank you very much for bringing this to our attention. Family 6 is now included in Figure 2, with panel A representing proteasome degradation and panel B for T-cell function. DOCK1/CRK/BCAR1 duplication is corrected.

7. In Supporting information please add title and Authors. Supplementary information for " title", Authors.

Authors’ response: Thank you for this suggestion. We have now added the title and author’s name to the supplementary information.

8. Please unify citation style in Supplementary Information.

Authors’ response: We have unified the citation style throughout the supplementary information to ensure consistency.

Reviewer 2 Report

The work is well organised and very useful. A possible but not mandatory suggestion is to better clarify which human proteins with SH3 domains are involved in diseases....The review deals with all kingdoms...and then it switches to humans....

Another issue is: are these data available? if yes do you have a browsable wen server? Not clear.

Author Response

The work is well organised and very useful. A possible but not mandatory suggestion is to better clarify which human proteins with SH3 domains are involved in diseases....The review deals with all kingdoms...and then it switches to humans....

Authors’ response: Thank you very much for your positive feedback on our work. We have included Table 2, which lists the names of the full SH3-containing proteins associated with each disease, thus increasing the clarity of our review. Regarding the scope of the review, we focus specifically on human SH3 domains. However, we mention in passing that SH3 domains occur in all five kingdoms of life to highlight their evolutionary significance.

Another issue is: are these data available? if yes do you have a browsable wen server? Not clear.

Authors’ response: Regarding data availability, all data mentioned in this paper are from the existing literature. While we do not have our own web server or database to search the data, readers can access the relevant information through a literature search by using the references provided in the main article and supplementary information.

Reviewer 3 Report

The manuscript “A Systematic Compilation of Human SH3 Domains: A Versatile 2 Superfamily in Cellular Signaling” by Mehrnaz Mehrabipour et al (REF: cells-2511053), reports a comprehensive review on one of the most widely found family of proteins (domains), the SH3 domains. I do not think there is such review in the literature and, thus, I believe this might be very important for researchers working in this field. Thus, I recommend this article for publication in Cells. However, a couple of minor things must be addressed before publication:

1.    The section Specificity of Binding must be divided into more subsections. It is hard to follow as is.

2.    In section Specificity of Binding, the authors do a good review on how the SH3 domains recognizes their respective targets. However, they are missing a very important point here. As reported by Palencia A et al. JBC 2010 (https://doi.org/10.1074/jbc.M109.048033) and later by Martin-Garcia et al. BJ. 2012 (https://doi.org/10.1042/BJ20111089), SH3 domains recognize their targets through a dual mechanism in which water molecules play a key role. This must also be considered and discussed in the manuscript.

3.    In the supplemental information, please remove the section on structural analysis. I do not think the authors have conducted any structural analysis on SH3 domains. All they show in figure 1 is known for decades. They have used PYMOL program to just visualize and make figure 1.

Manuscript is very well written and, perhaps, minor grammar edis might be needed. 

Author Response

The manuscript “A Systematic Compilation of Human SH3 Domains: A Versatile 2 Superfamily in Cellular Signaling” by Mehrnaz Mehrabipour et al (REF: cells-2511053), reports a comprehensive review on one of the most widely found family of proteins (domains), the SH3 domains. I do not think there is such review in the literature and, thus, I believe this might be very important for researchers working in this field. Thus, I recommend this article for publication in Cells. However, a couple of minor things must be addressed before publication:

1. The section Specificity of Binding must be divided into more subsections. It is hard to follow as is.

Authors’ response: Thank you very much for your positive feedback on our work. For clarity, we have included Table 1, which presents information on binding specificity in a clear tabular format.

2. In section Specificity of Binding, the authors do a good review on how the SH3 domains recognizes their respective targets. However, they are missing a very important point here. As reported by Palencia A et al. JBC 2010 (https://doi.org/10.1074/jbc.M109.048033) and later by Martin-Garcia et al. BJ. 2012 (https://doi.org/10.1042/BJ20111089), SH3 domains recognize their targets through a dual mechanism in which water molecules play a key role. This must also be considered and discussed in the manuscript.

Authors’ response: Thank you for raising this issue and indicating that all relevant published manuscripts need to be included in our article. We addressed this issue by including references 29-30 in lines 89-91 to highlight the importance of water molecules in the recognition process of SH3 domains versus PRMs.

3. In the supplemental information, please remove the section on structural analysis. I do not think the authors have conducted any structural analysis on SH3 domains. All they show in figure 1 is known for decades. They have used PYMOL program to just visualize and make figure 1.

Authors’ response: Thank you for highlighting this point. We apologize for the misleading information on structural analysis and have removed this section from the supplementary information.